# Label Distribution Propagation-based Label Completion for Crowdsourcing

**Tong Wu** [1]   **Liangxiao Jiang** [1]   **Wenjun Zhang** [1]   **Chaoqun Li** [2]

## Abstract

In real-world crowdsourcing scenarios, most workers often annotate a few instances only, which results in a significantly sparse crowdsourced label matrix and subsequently harms the performance of label integration algorithms. Recent work called worker similarity-based label completion (WSLC) has been proven to be an effective algorithm to addressing this issue. However, WSLC considers solely the correlation of the labels annotated by different workers on per individual instance while totally ignoring the correlation of the labels annotated by different workers among similar instances. To fill this gap, we propose a novel label distribution propagation-based label completion (LDPLC) algorithm. At first, we use worker similarity weighted majority voting to initialize a label distribution for each missing label. Then, we design a label distribution propagation algorithm to enable each missing label of each instance to iteratively absorb its neighbors' label distributions. Finally, we complete each missing label based on its converged label distribution. Experimental results on both real-world and simulated crowdsourced datasets show that LDPLC significantly outperforms WSLC in enhancing the performance of label integration algorithms. Our codes and datasets are available at https://github.com/jiangliangxiao/LDPLC.

## 1. Introduction

Supervised learning is a critical branch of machine learning, with the primary goal of learning the mapping between inputs and outputs from annotated data (Li & Tang, 2017; Jiang et al., 2019; Li et al., 2021). High-quality annotated data is crucial for the success of model training, but its acqui-

sition often presents significant challenges (Tu et al., 2019; Wu et al., 2023; Zhang et al., 2023). Traditionally, high-quality annotated data is generated through manual annotation by domain experts, but which is both time-consuming and expensive. As a result, efficiently obtaining large-scale and high-quality annotated data has become a major bottleneck in the practical application of supervised learning.

To break through this bottleneck, crowdsourcing is introduced as an efficient and cost-effective method to rapidly obtain large volumes of annotated data (Xu et al., 2022; Ji et al., 2023) . Currently, mainstream crowdsourcing platforms include Amazon Mechanical Turk (AMT), CloudCrowd and CrowdFlower. These platforms utilize the internet to outsource tasks traditionally performed by domain experts to a distributed group of non-professional crowd workers (Liang et al., 2023; Mak & Lam, 2023). However, due to variations in the knowledge and abilities of crowd workers, the annotated labels are often accompanied by noise and inconsistent quality. To address this issue, each instance is typically assigned to multiple crowd workers and thus obtains its multiple noisy label set. Then, label integration is used to infer the integrated label of each instance from its multiple noisy label set (Sheng et al., 2008; Zhang et al., 2024a). In recent years, label integration has attracted widespread attention from researchers, and a large number of algorithms have been proposed from different perspectives (Zheng et al., 2017; Zhang, 2022; Zhang et al., 2024c).

However, in real-world crowdsourcing scenarios, most workers often annotate a few instances only, thus most instances do not receive sufficient labels, leading to a large number of missing labels in the crowdsourced label matrix. The sparse label matrix significantly harms the performance of label integration algorithms. To mitigate the problem of sparsity, a small number of researchers have paid attention to the importance of the preprocessing of crowdsourced label matrix and proposed a few label completion algorithms. For example, to address binary-class crowdsourcing tasks, Jung & Lease (2012) propose a probabilistic matrix factorization (PMF)-based label completion algorithm. To further improve its performance, Yang et al. (2024) propose a PMF-based three-stage label completion (PMF-TLC) algorithm. Recently, to address multi-class crowdsourcing tasks, Wu et al. (2024) propose worker similarity-based label completion (WSLC), which significantly improves the performance

[1]School of Computer Science, China University of Geosciences, Wuhan 430074, China [2]School of Mathematics and Physics, China University of Geosciences, Wuhan 430074, China. Correspondence to: Liangxiao Jiang <ljiang@cug.edu.cn>.

*Proceedings of the 42nd International Conference on Machine Learning*, Vancouver, Canada. PMLR 267, 2025. Copyright 2025 by the author(s).

of label integration algorithms.

To the best of our knowledge, WSLC considers solely the correlation of the labels annotated by different workers on per individual instance while totally ignoring the correlation of the labels annotated by different workers among similar instances. To fill this gap, we propose a novel label completion algorithm called label distribution propagation-based label completion (LDPLC). At first, we use worker similarity weighted majority voting to initialize a label distribution for each missing label. Then, we design a label distribution propagation algorithm to enable each missing label of each instance to iteratively absorb its neighbors' label distributions. Finally, we complete each missing label based on its converged label distribution. In general, the main contributions of this work can be summarized as follows:

- We find that WSLC considers solely the correlation of the labels annotated by different workers on per individual instance while totally ignoring the correlation of the labels annotated by different workers among similar instances.

- We propose a label distribution propagation-based label completion (LDPLC) algorithm. In LDPLC, we use worker similarity weighted majority voting to initialize a label distribution for each missing label and then design a label distribution propagation algorithm to enable each missing label of each instance to iteratively absorb its neighbors' label distributions.

- We conduct a series of experiments on both real-world and simulated crowdsourced datasets. The experimental results indicate that LDPLC further improves the performance of label integration algorithms compared to WSLC.

The rest of the paper is organized as follows. Section 2 briefly reviews related work on label integration and label completion. Section 3 proposes our LDPLC. Section 4 reports the experimental setup and results. Section 5 concludes the paper and outlines future work.

## 2. Related Work

Over the past few years, a large number of label integration algorithms have been proposed from different perspectives. Among them, majority voting (MV) is a simple yet effective label integration algorithm, which selects the most frequently assigned label as the integrated label. Since MV ignores the annotation quality of each crowd worker, it is sensitive to noise and unsuitable for complex tasks. To improve its performance, researchers propose many more sophisticated label integration algorithms. For example, Dawid & Skene (1979) propose Dawid-Skene (DS), which

utilizes the expectation maximization framework to iteratively estimate true labels and observer error rates. Demartini et al. (2012) propose ZenCrowd(ZC), which employs a two-element probability parameter to iteratively estimate the reliability of each individual worker. Li & Yu (2014) proposes iterative weighted majority voting (IWMV), which optimizes error rate bounds to find reliable workers. Zhang et al. (2015b) propose positive label frequency threshold (PLAT), which is designed to address imbalanced scenarios and uses multiple noisy label sets to model the decision thresholds. Zhang et al. (2016) also propose ground truth inference using clustering (GTIC), which generates features from multiple noisy label sets and then clusters instances into different groups utilizing generated features. Tao et al. (2021) propose differential evolution-based weighted soft majority voting (DEWSMV), which exploits a differential evolution algorithm to estimate the quality of crowd workers labeling different instances. Chen et al. (2022) propose label augmented and weighted majority voting (LAWMV), which merges the neighbors' multiple noisy label sets to obtain its augmented multiple noisy label set and weights each neighbor. Jiang et al. (2022) propose multiple noisy label distribution propagation (MNLDP), which at first estimates the multiple noisy label distribution of each instance from its multiple noisy label set and then propagates its multiple noisy label distribution to its nearest neighbors.

To the best of our knowledge, all above works primarily aim to design more effective label integration algorithms while paying little attention to the preprocessing of crowdsourced label matrix before label integration. In real-world crowdsourcing scenarios, most workers often annotate a few instances only, which results in a significantly sparse crowdsourced label matrix and subsequently harms the performance of label integration algorithms. To mitigate the problem of sparsity, a small number of researchers have paid attention to the importance of the preprocessing of crowdsourced label matrix and proposed a few label completion algorithms. For example, Jung & Lease (2012) propose an algorithm to utilize probabilistic matrix factorization (PMF) to address the challenge of predicting missing labels in binary-class crowdsourcing tasks. To further enhance its performance, Yang et al. (2024) propose a PMF-based three-stage label completion (PMF-TLC) algorithm, which flips low-quality raw labels based on a confidence-based strategy, employs PMF to complete missing labels, and finally filters low-quality output labels using a between-class margin-based algorithm. Recently, to address multi-class crowdsourcing tasks, Wu et al. (2024) propose worker similarity-based label completion (WSLC), which completes each missing label by worker similarity. Specifically, they estimate the similarity between each pair of workers and complete the missing labels based on the estimated worker similarity and the labels from similar workers. However, WSLC

considers solely the correlation of the labels annotated by different workers on per individual instance while totally ignoring the correlation of the labels annotated by different workers among similar instances. To fill this gap, we propose a novel label distribution propagation-based label completion (LDPLC) algorithm in Section 3.

## 3. The Proposed LDPLC

### 3.1. Motivation

As we discussed in Section 2, existing works have primarily focused on label integration while paying little attention to the preprocessing of crowdsourced datasets. In real-world crowdsourcing scenarios, most workers often annotate a few instances only, which results in a significantly sparse crowdsourced label matrix. WSLC has been validated as an effective label completion algorithm to address this issue. WSLC assumes that different workers with similar cognitive abilities will annotate similar labels on the same instance. Based on this assumption, WSLC completes the missing labels based on the worker similarity and the labels from similar workers. However, since each instance is annotated by only a few workers, WSLC relies on limited information from these similar workers to complete missing labels, which does not fully use the correlation of the labels in crowdsourced dataset. To overcome this limitation, we introduce the assumption that the same workers will also annotate similar labels on similar instances. This additional perspective allows each missing label to not only absorb the label distribution information from similar workers on the corresponding instance but also absorb the label distribution information from the corresponding worker across similar neighboring instances.

Specifically, we attempt to initialize a label distribution for each missing label and then design a label distribution propagation algorithm to enable each missing label to absorb information from its neighbors. Therefore, two core issues need to be addressed. The first issue is how to initialize a label distribution. WSLC assumes that different workers with similar cognitive abilities will annotate similar labels on the same instance. Based on this assumption, we use worker similarity weighted majority voting to initialize a label distribution for each missing label. The second issue is how to propagate the initialized label distribution. We assume that the same worker will also annotate similar labels on similar instances. Therefore, we first query neighbors, then optimize their weights, and finally propagate the distribution from weighted neighbors to each missing label.

To this end, we propose a novel label distribution propagation-based label completion (LDPLC) algorithm. Its framework can be graphically shown in Figure 1. Firstly, we use Pearson correlation to learn a feature vector for

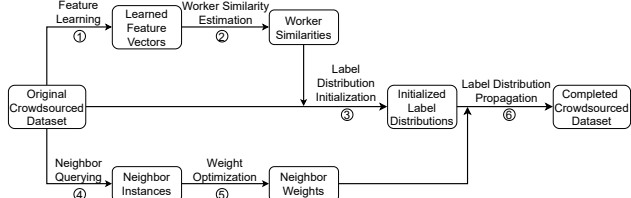

*Figure 1.* Overall framework of LDPLC.

each worker. Secondly, we use cosine similarity to estimate worker similarity for each pair of workers based on the learned feature vectors. Thirdly, we initialize a label distribution for each missing label based on the original crowdsourced dataset and the estimated worker similarity of each pair of workers. Fourthly, we query neighbors for each instance in the feature space of the original crowdsourced dataset. Fifthly, we optimize the neighbors' weights using local linear embedding. Finally, we propagate the initialized label distribution from weighted neighbors to each missing label of each instance and then complete each missing label based on its converged label distribution. In the following subsections, we will present the detailed process.

### 3.2. Label Distribution Initialization

In a crowdsourcing scenario, a crowdsourced dataset $D$ can be denoted as $\{(\boldsymbol{x}_i, \boldsymbol{L}_i)\}_{i=1}^{N}$, where $N$ denotes the number of instances. $\boldsymbol{x}_i$ donates the $i$-th instance in $D$, which can be denoted as $\{a_{i1}, \ldots, a_{im}, \ldots, a_{iM}\}$. $M$ denotes the number of features and $a_{im}$ denotes the $m$-th feature value of the $i$-th instance. $\boldsymbol{L}_i$ denotes the multiple noisy label set of $\boldsymbol{x}_i$, which is denoted as $\{l_{i1}, \ldots, l_{ir}, \ldots, l_{iR}\}$. $R$ denotes the number of workers and $l_{ir}$ denotes the label annotated by $r$-th worker $u_r$ on $\boldsymbol{x}_i$. $l_{ir}$ is drawn from a fixed set $\{-1, c_1, \ldots, c_q, \ldots, c_Q\}$, where $Q$ denotes the number of classes, $c_q$ denotes the $q$-th class and $-1$ means that $u_r$ does not annotate $\boldsymbol{x}_i$.

According to the framework of LDPLC, worker similarity estimation is a crucial step in label distribution initialization. In worker similarity estimation, we first need to construct a dataset $D_r$ for $u_r$, where $D_r$ consists of all the instances annotated by $u_r$ in $D$. Next, we learn a feature vector $\boldsymbol{V}_r$ for $u_r$ based on the label variable and the original feature variables of $D_r$. We denote $\boldsymbol{V}_r$ as $\{v_{r1}, \ldots, v_{rm}, \ldots, v_{rM}\}$, where $v_{rm}$ denotes the $m$-th feature value of $u_r$. Here, we use Pearson correlation to learn the feature vector (Hall, 2000; Wu et al., 2024) by Eq. (1).

$$cor(\boldsymbol{C}_r, \boldsymbol{A}_{rm})$$
$$= \frac{\sum_{i=1}^{|D_r|}(l_{ir} - \bar{C}_r)(a_{irm} - \bar{A}_{rm})}{\sqrt{\sum_{i=1}^{|D_r|}(l_{ir} - \bar{C}_r)^2}\sqrt{\sum_{i=1}^{|D_r|}(a_{irm} - \bar{A}_{rm})^2}}, \quad (1)$$

where $\boldsymbol{C}_r$ and $\boldsymbol{A}_{rm}$ denote the label variable and the $m$-th

feature variable of $D_r$, respectively. And $a_{irm}$ denotes the $m$-th feature value of the $i$-th instance in $D_r$. $\bar{C}_r$ and $\bar{A}_{rm}$ denote the mean values of $C_r$ and $A_{rm}$, respectively. $\bar{C}_r$ and $\bar{A}_{rm}$ can be calculated by Eqs. (2) and (3), respectively.

$$\bar{C}_r = \frac{1}{|D_r|} \sum_{i=1}^{|D_r|} l_{ir}. \tag{2}$$

$$\bar{A}_{rm} = \frac{1}{|D_r|} \sum_{i=1}^{|D_r|} a_{irm}. \tag{3}$$

Since the label variable is a discrete variable, we use weighted Pearson correlation. To be specific, if we want to calculate Pearson correlation between $C_r$ and $A_{rm}$, we can decompose $C_r$ into multiple binary label variables, each binary label variable corresponding to a value of $C_r$. The value of each binary label variable takes value 1 when the corresponding value of $C_r$ occurs and 0 for other values. Next, we calculate Pearson correlation between each binary label variable and $A_{rm}$, and perform a weighted summation by the prior probability of the corresponding value. If $A_{rm}$ is also discrete, we decompose it into multiple binary feature variables and then calculate Pearson correlation between each binary label variable of $C_r$ and each binary feature variable of $A_{rm}$. Based on the above analysis, the feature value $v_{rm}$ of $u_r$ is calculated by Eq. (4).

$$v_{rm} = \begin{cases} \sum_{l_r} p(C_r = l_r) \, cor(C_{br}, A_{rm}), \\ \qquad\qquad if\ A_{rm}\ is\ a\ continuous\ variable, \\ \sum_{l_r} \sum_{a_{rm}} p(C_r = l_r, A_{rm} = a_{rm}) \, cor(C_{br}, A_{brm}), \\ \qquad\qquad\qquad\qquad\qquad\qquad otherwise. \end{cases} \tag{4}$$

where $l_r$ and $a_{rm}$ denotes the values that $C_r$ and $A_{rm}$ take, respectively. $C_{br}$ and $A_{brm}$ are a binary label variable of $C_r$ and a binary feature variable of $A_{rm}$, respectively. The prior probabilities $p(C_r = l_r)$ and $p(C_r = l_r, A_{rm} = a_{rm})$ can be calculated by Eqs. (5) and (6), respectively.

$$p(C_r = l_r) = \frac{\sum_{i=1}^{|D_r|} \delta(l_{ir}, l_r)}{|D_r|}, \tag{5}$$

$$p(C_r = l_r, A_{rm} = a_{rm}) = \frac{\sum_{i=1}^{|D_r|} \delta(l_{ir}, l_r)\delta(a_{irm}, a_{rm})}{|D_r|}, \tag{6}$$

where $\delta(.)$ denotes an indicator function that outputs 1 if the two input values are equal and outputs 0 otherwise.

By Eqs. (1) - (6), we calculate the learned feature vector $V_r$ for $u_r$. Finally, we use cosine similarity to estimate worker similarity $s(u_r, u_{r'})$ between $u_r$ and $u_{r'}$ based on $V_r$ and

$V_{r'}$, which is calculated by Eq. (7).

$$cos(V_r, V_{r'}) = \frac{V_r \cdot V_{r'}}{|V_r||V_{r'}|}$$
$$= \frac{\sum_{m=1}^{M} v_{rm} v_{r'm}}{\sqrt{\sum_{m=1}^{M} v_{rm}^2} \sqrt{\sum_{m=1}^{M} v_{r'm}^2}}. \tag{7}$$

Since $cos(V_r, V_{r'})$ takes values in the interval [-1,1], we take max-min normalization for the values of $cos(V_r, V_{r'})$, which is calculated by Eq. (8).

$$s(u_r, u_{r'}) = \frac{cos(V_r, V_{r'}) - (-1)}{1 - (-1)}. \tag{8}$$

With the above calculation of worker similarity, we can proceed to initialize a label distribution for each label. Let $P_{ir} = \{p_{ir1}, \ldots, p_{irq}, \ldots, p_{irQ}\}$ denote the label distribution of $l_{ir}$, where $p_{irq}$ denotes the probability $l_{ir}$ belongs to $c_q$. Therefore, for annotated labels and unannotated labels, we can directly initialize their label distributions by Eqs. (9) and (10), respectively.

$$p_{irq} = \begin{cases} 1, \ if\ l_{ir} == c_q \\ 0, \ otherwise \end{cases}. \tag{9}$$

$$p_{irq} = \frac{\sum_{r'=1}^{R} \delta(l_{ir'}, c_q) s(u_r, u_{r'})}{\sum_{q=1}^{Q} \sum_{r'=1}^{R} \delta(l_{ir'}, c_q) s(u_r, u_{r'})}. \tag{10}$$

### 3.3. Label Distribution Propagation

In the process of label distribution propagation, we design an algorithm to enable each label to iteratively absorb information from its neighbors. First, we use the HEOM metric to query $K$ neighbors of $x_i$, where neighbors are denoted as $\mathcal{N}_i = \{x_{ik}\}_{k=1}^{K}$ and $x_{ik}$ denotes $k$-th neighbor of $x_i$. Then, we use the local linear embedding (LLE) to optimize the weights of $K$ neighbors (Miao et al., 2022; Ghojogh et al., 2022), where the weights are denoted as $w_i = \{w_{ik}\}_{k=1}^{K}$. Here, we construct a minimization objective function based on LLE as Eq. (11).

$$L(w_i) = \sum_{k_1, k_2 : x_{k_1}, x_{k_2} \in \mathcal{N}_i} w_{ik_1} (x_i - x_{k_1})^T (x_i - x_{k_2}) w_{ik_2}. \tag{11}$$

This objective can be solved using the least squares algorithm, which is defined as Eq. (12).

$$\min_{w_i} L(w_i)$$
$$s.t. \begin{cases} \sum_{k=1}^{K} w_{ik} = 1 \\ \forall w_{ik} \in w_i, w_{ik} \geq 0 \end{cases}. \tag{12}$$

**Algorithm 1** LDPLC

1: **Input**: $D = \{(\boldsymbol{x}_i, \boldsymbol{L}_i)\}_{i=1}^{N}$ -a crowdsourced dataset
2: **Output**: $D'$-the completed crowdsourced dataset
3: **for** $r = 1$ to $R$ **do**
4:     Construct a dataset $D_r$ for $u_r$
5:     **for** $m = 1$ to $M$ **do**
6:         Learn the feature value $v_{rm}$ of $u_r$ by Eqs. (1)-(6)
7:     **end for**
8: **end for**
9: **for** $r = 1$ to $R$ **do**
10:     **for** $r' = 1$ to $R$ **do**
11:         Estimate the worker similarity $s(u_r, u_{r'})$ by Eqs. (7) and (8)
12:     **end for**
13: **end for**
14: **for** $i = 1$ to $N$ **do**
15:     **for** $r = 1$ to $R$ **do**
16:         **if** $l_{ir}! = -1$ **then**
17:             Initialize the label distribution $\boldsymbol{P}_{ir}$ by Eq. (9)
18:         **else**
19:             Initialize the label distribution $\boldsymbol{P}_{ir}$ by Eq. (10)
20:         **end if**
21:     **end for**
22: **end for**
23: **for** $i = 1$ to $N$ **do**
24:     Query the neighbors $\mathcal{N}_i$ for $\boldsymbol{x}_i$
25:     Optimize the neighbors' weights $\boldsymbol{w}_i$ by Eqs. (11) and (12)
26: **end for**
27: **for** $t = 1$ to $T$ **do**
28:     **for** $i = 1$ to $N$ **do**
29:         **for** $r = 1$ to $R$ **do**
30:             Propagate the neighbors' label distributions to $\boldsymbol{P}_{ir}$ by Eq. (13)
31:         **end for**
32:     **end for**
33: **end for**
34: **for** $i = 1$ to $N$ **do**
35:     **for** $r = 1$ to $R$ **do**
36:         **if** $l_{ir} == -1$ **then**
37:             Complete the missing label $l_{ir}$ by Eq. (14)
38:         **end if**
39:     **end for**
40: **end for**
41: **return** $D'$

Therefore, we can obtain a weight $w_{ik}$ for each neighbor $\boldsymbol{x}_{ik}$ of $\boldsymbol{x}_i$. Next, we propagate the label distributions from each neighbor $\boldsymbol{x}_{ik}$ to $\boldsymbol{x}_i$ by Eq. (13).

$$\boldsymbol{P}_{ir}^{t+1} = \frac{\sum_{k:x_{ik} \in \mathcal{N}_i} w_{ik} \boldsymbol{P}_{irk}^t + \boldsymbol{P}_{ir}}{2}, \tag{13}$$

where $\boldsymbol{P}_{irk}^t$ denotes the distribution of the label annotated by $u_r$ on $x_{ik}$ after t iterations of propagation. When the propagation converges, we obtain the converged distribution $\boldsymbol{P}_{ir}^*$ of each missing label $l_{ir}$. Finally, we complete $l_{ir}$ based on $\boldsymbol{P}_{ir}^*$, which is calculated by Eq. (14).

$$l_{ir} = \underset{c_q \in \{c_1, c_2, \ldots, c_Q\}}{arg\,max} \boldsymbol{P}_{ir}^* . \tag{14}$$

To this end, the learning algorithm of LDPLC is described

in **Algorithm** 1. After the above process, the completed $D'$ can be fed into existing label integration algorithms to enhance their performance.

### 3.4. Time Complexity Analysis

In **Algorithm** 1, lines 3-8 learn feature vectors with a time complexity of $O(R(N + M(n_l n_a |D_r|)))$, where $n_l$ and $n_a$ are the average number of values for a label variable and an original feature variable, respectively. Lines 9-13 estimate worker similarity with a time complexity of $O(R^2 M)$. Lines 14-22 initialize label distributions with a time complexity of $O(NR(Q + R))$. Lines 23-26 identify neighbors and optimize their weights with a time complexity of $O(N^2 M + NK^3)$. Lines 27-33 propagate the label distributions with a time complexity of $O(TNRKQ)$. Finally, lines 34-40 complete missing labels with a time complexity of $O(NRQ)$. If only the highest order terms are taken, the overall time complexity of LDPLC is $O(n_l n_a |D_r| RM + R^2 M + NR^2 + N^2 M + NK^3 + TNRKQ)$.

### 3.5. Convergence Analysis

In the part of the previous subsection, we describe the label distribution propagation for $u_r$ on $\boldsymbol{x}_i$. Now, we discuss the propagation process of $u_r$ across the whole dataset $D$. First, we define $\mathcal{P}_r = [\boldsymbol{P}_{1r}^T, \ldots, \boldsymbol{P}_{ir}^T, \ldots, \boldsymbol{P}_{Nr}^T]^T$, which has a size of $N \times Q$. The matrix $\mathcal{W}$ is defined as the weights of the nearest neighbors, with a size of $N \times N$, where each element $\mathcal{W}_{ij}$ of $\mathcal{W}$ is denoted by Eq. (15).

$$\mathcal{W}_{ij} = \begin{cases} w_{ik}, \ if \ x_j == x_{ik} \\ 0, \ otherwise \end{cases} . \tag{15}$$

Then, we update Eq. (13) to Eq. (16).

$$\mathcal{P}_r^{t+1} = \frac{\mathcal{W}}{2} \mathcal{P}_r^t + \frac{\mathcal{P}_r}{2}. \tag{16}$$

Eq. (16) is a linear iterative formula, where $\frac{\mathcal{W}}{2}$ controls the weight of propagation, and $\frac{\mathcal{P}_r}{2}$ preserves the influence of the original distribution. When $t$ is equal to 0, it is obtained that $\mathcal{P}_r^0$ is equal to $\mathcal{P}_r$. By expanding the recursion using mathematical induction, we derive the explicit expression of $\mathcal{P}_r^t$ as Eq. (17).

$$\mathcal{P}_r^t = (\frac{\mathcal{W}}{2})^t \mathcal{P}_r + \sum_{i=0}^{t-1} (\frac{\mathcal{W}}{2})^i \frac{\mathcal{P}_r}{2}. \tag{17}$$

Since the elements in $\mathcal{W}$ satisfy $\mathcal{W}_{ij} \in [0, 1]$, the spectral radius of $\frac{\mathcal{W}}{2}$ is less than 1, ensuring that $(\frac{\mathcal{W}}{2})^t$ approaches 0 as $t$ increases, as shown in Eq. (18),

$$\lim_{t \to \infty} (\frac{\mathcal{W}}{2})^t = \boldsymbol{0}. \tag{18}$$

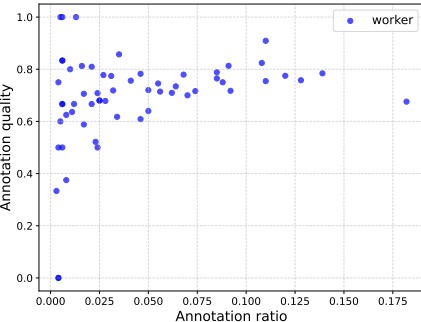

Figure 2. Worker distribution of the "LabelMe" dataset.

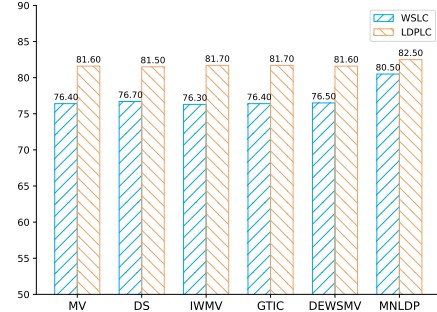

Figure 3. Integration accuracy (%) comparisons of six label integration algorithms after label completion by WSLC and LDPLC on the "LabelMe" dataset, respectively.

According to the geometric series formula, $\sum_{i=0}^{t-1}(\frac{\boldsymbol{\mathcal{W}}}{2})^i$ can be calculated by Eq. (19).

$$\lim_{t \to \infty} \sum_{i=0}^{t-1}(\frac{\boldsymbol{\mathcal{W}}}{2})^i = \lim_{t \to \infty} \frac{(\frac{\boldsymbol{\mathcal{W}}}{2})^0 - (\frac{\boldsymbol{\mathcal{W}}}{2})^t}{1 - \frac{\boldsymbol{\mathcal{W}}}{2}}$$
$$= (1 - \frac{\boldsymbol{\mathcal{W}}}{2})^{-1}. \tag{19}$$

Therefore, $\boldsymbol{\mathcal{P}}_r^t$ converges to a fixed matrix, which is calculated by Eq. (20).

$$\lim_{t \to \infty} \boldsymbol{\mathcal{P}}_r^t = (1 - \frac{\boldsymbol{\mathcal{W}}}{2})^{-1} \frac{\boldsymbol{\mathcal{P}}_r}{2}. \tag{20}$$

## 4. Experiments and Results

### 4.1. Experiments Setup

To validate the effectiveness of the LDPLC algorithm, we use WSLC (Wu et al., 2024) and LDPLC to complete missing labels in real-world and simulated datasets, respectively, and compare their effectiveness in terms of the integration accuracy of each label integration algorithm. In our experiments, we implement LDPLC and WSLC on the Crowd Environment and its Knowledge Analysis (CEKA) (Zhang et al., 2015a) platform. The label integration algorithms used include MV (Sheng et al., 2008), DS (Dawid & Skene, 1979), IWMV (Li & Yu, 2014), GTIC (Zhang et al., 2016), DEWSMV (Tao et al., 2021) and MNLDP (Jiang et al., 2022). For MV and GTIC, we use the existing implementations on the CEKA platform. For DEWSMV and MNLDP, we implement them on the CEKA platform. The parameter settings of all label integration algorithms are consistent with those specified in their original papers. For simplicity, we set both the number of neighbors $K$ and the number of iterations $T$ to 5. All experiments are independently repeated ten times, and the average results of the ten experiments are used as the final results.

### 4.2. Experiments on Real-world Datasets

To evaluate the performance of LDPLC, we conduct experiments on the widely used real-word crowdsourced dataset "LabelMe". The dataset is collected from AMT platform, designed for image classification. The dataset "LabelMe" contains 1000 instances, and includes 8 classes: "highway", "inside city", "tall building", "street", "forest", "coast", "mountain", and "open country". In total, there are 2547 multiple noisy labels from 59 workers, with a mean annotation quality of 74.05% and a mean annotation ratio of 4.32%. Each worker's annotation ratio and annotation quality denote the proportions of the instances he annotated in all instances and the true labels in all his annotated labels, respectively, shown in Figure 2. It can be seen that most workers' annotation qualities are greater than 0.5, enabling each label to absorb more accurate information during the propagation process. Although a few workers' annotation qualities are below 0.5, their annotation ratios are very low, resulting in minimal impact on the propagation process. Therefore, we use this dataset in our experiments.

Figure 3 shows the integration accuracies of six label integration algorithms after label completion by WSLC and LDPLC on the "LabelMe" dataset, respectively. We can see that after label completion using LDPLC, the integration accuracy of each integration algorithm improves significantly. Specifically, the integration accuracies of MV (81.60%), DS (81.50%), IWMV (81.70%), GTIC (81.70%), DEWSMV (81.60%) and MNLDP (82.50%) after label completion by LDPLC are much higher than those of MV (76.40%), DS (76.70%), IWMV (76.30%), GTIC (76.40%), DEWSMV (76.50%) and MNLDP (80.50%) after label completion by WSLC, respectively. From these results, we can conclude that LDPLC further improves the performance of all label integration algorithms compared to WSLC. Except for the dataset "LabelMe", we also conduct experiments on two other widely used real-world crowdsourced datasets "Ruters" and "Leaves" to validate the effectiveness of LDPLC. Due to

the limited pages, the detailed description of these datasets and experimental results are provided in **Appendix A**. These results also demonstrate the effectiveness of LDPLC.

### 4.3. Experiments on Simulated Datasets

In this subsection, we conduct a series of experiments on the whole 34 simulated datasets published on the CEKA platform. The detailed description of these datasets is provided in Table 1. Here, "#Instances" denotes the number of instances, "#Features" denotes the number of attributes, "#Classes" denotes the number of classes, "Missing" denotes whether the dataset contains missing values and "Feature type" denotes the type of attributes the dataset contains. Since MNLDP cannot handle missing feature values, we need to complete missing feature values on the datasets. Specifically, we replace all missing feature values with the mean of numerical feature values or the modes of nominal feature values from the available data, using an unsupervised feature filter ReplaceMissingValues in the Waikato Environment and Knowledge Analysis (WEKA) (Witten et al., 2011) platform. Subsequently, we simulate 40 crowd workers to replicate a real-world crowdsourcing process according to the characteristics of the "LabelMe" dataset. Observed in Figure 2, most workers' annotation ratios and annotation qualities are in the interval $[0, 0.1]$ and $[0.6, 0.9]$, respectively. Therefore, for each simulated crowd worker $u_r$, we randomly sample its annotation ratio and annotation quality from the uniform distribution $[0, 0.1]$ and $[0.6, 0.9]$, respectively. As a result, approximately 95% of the labels in the label matrix are missing, reflecting the sparsity observed in real-world crowdsourced datasets.

Table 2 presents the integration accuracy comparisons of six label integration algorithms after label completion by WSLC and LDPLC, respectively. Based on these results, we compare each pair of label integration algorithms after label completion by WSLC and LDPLC using the corrected paired two-tailed t-test with the significance level $\alpha = 0.05$ (Nadeau & Bengio, 2003; Zhang et al., 2024b). The symbols ● and ○ in the table indicate that the integration accuracy has a statistically significant improvement or degradation using our proposed LDPLC compared to WSLC, respectively. The bottom of Table 2 denotes the mean integration accuracy and the $Win/Tie/Lose$ ($W/T/L$) across the 34 datasets. The mean integration accuracy reflects the average performance of different label integration algorithms after label completion by WSLC and LDPLC. The $W/T/L$ indicates that in terms of improving the performance of label integration algorithms, LDPLC wins WSLC on $W$ datasets, ties on $T$ datasets, and loses on $L$ datasets. These experimental results verify the effectiveness of our proposed LDPLC. Specific conclusions are summarized as follows:

- The average integration accuracies of MV (83.51%),

*Table 1.* Description of 34 simulated datasets.

| Dataset | #Instances | #Features | #Classes | Missing | Feature type |
|---|---|---|---|---|---|
| anneal | 898 | 38 | 6 | yes | hybrid |
| audiology | 226 | 69 | 24 | yes | nominal |
| autos | 205 | 25 | 7 | yes | hybrid |
| balance-scale | 625 | 4 | 3 | no | numeric |
| biodeg | 1055 | 41 | 2 | no | numeric |
| breast-cancer | 286 | 9 | 2 | yes | nominal |
| breast-w | 699 | 9 | 2 | yes | numeric |
| car | 1728 | 6 | 4 | no | nominal |
| credit-a | 690 | 15 | 2 | yes | hybrid |
| credit-g | 1000 | 20 | 2 | no | hybrid |
| diabetes | 768 | 8 | 2 | no | numeric |
| heart-c | 303 | 13 | 5 | yes | hybrid |
| heart-h | 294 | 13 | 5 | yes | hybrid |
| heart-statlog | 270 | 13 | 2 | no | numeric |
| hepatitis | 155 | 19 | 2 | yes | hybrid |
| horse-colic | 368 | 22 | 2 | yes | hybrid |
| hypothyroid | 3772 | 29 | 4 | yes | hybrid |
| ionosphere | 351 | 34 | 2 | no | numeric |
| iris | 150 | 4 | 3 | no | numeric |
| kr-vs-kp | 3196 | 36 | 2 | no | nominal |
| labor | 57 | 16 | 2 | yes | hybrid |
| letter | 20000 | 16 | 26 | no | numeric |
| lymph | 148 | 18 | 4 | no | hybrid |
| mushroom | 8124 | 22 | 2 | yes | nominal |
| segment | 2310 | 19 | 7 | no | numeric |
| sick | 3772 | 29 | 2 | yes | hybrid |
| sonar | 208 | 60 | 2 | no | numeric |
| spambase | 4601 | 57 | 2 | no | numeric |
| tic-tac-toe | 958 | 9 | 2 | no | nominal |
| vehicle | 846 | 18 | 4 | no | numeric |
| vote | 435 | 16 | 2 | yes | nominal |
| vowel | 990 | 13 | 11 | no | hybrid |
| waveform | 5000 | 40 | 3 | no | numeric |
| zoo | 101 | 17 | 7 | no | hybrid |

DS (83.57%), IWMV (83.54%), GTIC (83.45%), DEWSMV (83.54%) and MNLDP (83.99%) after label completion by LDPLC are much higher than those of MV (74.56%), DS (74.08%), IWMV (74.57%), GTIC (74.43%), DEWSMV (74.55%) and MNLDP (76.80%) after label completion by WSLC, respectively. The result once again demonstrates that LDPLC further improves the performance of all label integration algorithms compared to WSLC.

- LDPLC significantly wins WSLC on 30, 30, 30, 30, 30, and 27 datasets for MV, DS, IWMV, GTIC, DEWSMV and MNLDP, respectively, while significantly losing only on 1 dataset. In t-test results, the number of datasets in which LDPLC wins significantly ($W$) is always much higher than the number of datasets in which it loses significantly ($L$) for all label integration algorithms. This further strongly demonstrates the effectiveness of LDPLC.

In addition, to validate the robustness of LDPLC under different annotation quality distributions, we conduct another set of experiments. In new experiments, we replace the distribution of the annotation quality with a Gaussian distribution $N(0.75, 0.15^2)$, while keeping other settings

*Table 2.* Integration accuracy (%) comparisons of six label integration algorithms after label completion by WSLC and LDPLC on the uniform distribution, respectively.

| Dataset | MV WSLC | MV LDPLC | DS WSLC | DS LDPLC | IWMV WSLC | IWMV LDPLC | GTIC WSLC | GTIC LDPLC | DEWSMV WSLC | DEWSMV LDPLC | MNLDP WSLC | MNLDP LDPLC |
|---|---|---|---|---|---|---|---|---|---|---|---|---|
| anneal | 70.45 | 86.31 ● | 68.21 | 86.24 ● | 70.41 | 86.29 ● | 69.82 | 86.26 ● | 70.35 | 86.31 ● | 74.70 | 86.38 ● |
| audiology | 69.25 | 81.19 ● | 69.20 | 81.24 ● | 69.11 | 81.33 ● | 68.98 | 80.44 ● | 69.12 | 81.19 ● | 71.20 | 80.62 ● |
| autos | 71.76 | 82.68 ● | 71.61 | 82.68 ● | 71.81 | 82.63 ● | 71.85 | 81.95 ● | 71.66 | 82.68 ● | 73.95 | 81.17 ● |
| balance-scale | 75.15 | 83.46 ● | 72.34 | 83.01 ● | 74.83 | 83.34 ● | 73.46 | 83.34 ● | 74.99 | 83.44 ● | 79.49 | 84.16 ● |
| biodeg | 77.95 | 81.42 ● | 77.72 | 81.31 ● | 77.92 | 81.39 ● | 77.82 | 81.44 ● | 77.92 | 81.44 ● | 79.99 | 81.75 ● |
| breast-cancer | 78.39 | 78.15 | 77.10 | 78.14 | 78.04 | 78.14 | 77.80 | 78.11 | 78.50 | 78.32 | 79.27 | 77.72 |
| breast-w | 78.48 | 85.81 ● | 78.37 | 85.64 ● | 78.44 | 85.72 ● | 78.43 | 85.84 ● | 78.44 | 85.82 ● | 82.47 | 88.58 ● |
| car | 78.40 | 86.24 ● | 75.71 | 86.25 ● | 78.41 | 86.25 ● | 77.64 | 86.25 ● | 78.41 | 86.25 ● | 83.81 | 86.49 ● |
| credit-a | 74.59 | 81.06 ● | 74.75 | 81.03 ● | 74.65 | 81.05 ● | 74.64 | 81.06 ● | 74.59 | 81.06 ● | 76.42 | 81.32 ● |
| credit-g | 77.00 | 77.80 | 75.95 | 77.72 | 76.76 | 77.79 | 76.61 | 77.79 | 76.93 | 77.82 | 78.44 | 77.55 |
| diabetes | 76.63 | 78.31 | 76.05 | 78.09 | 76.56 | 78.15 | 76.39 | 78.27 | 76.69 | 78.27 | 78.16 | 78.18 |
| heart-c | 77.33 | 84.29 ● | 76.27 | 84.23 ● | 77.16 | 84.19 ● | 75.58 | 84.33 ● | 77.43 | 84.33 ● | 80.69 | 84.16 ● |
| heart-h | 77.18 | 83.74 ● | 76.67 | 83.67 ● | 77.07 | 83.70 ● | 76.33 | 83.64 ● | 77.14 | 83.70 ● | 80.54 | 83.23 |
| heart-statlog | 74.92 | 78.52 ● | 74.48 | 78.59 ● | 74.44 | 78.81 ● | 74.37 | 78.41 ● | 74.96 | 78.56 ● | 74.26 | 78.85 ● |
| hepatitis | 69.36 | 83.03 ● | 69.74 | 82.84 ● | 71.29 | 83.03 ● | 70.26 | 82.84 ● | 69.29 | 83.03 ● | 67.55 | 83.94 ● |
| horse-colic | 72.31 | 77.69 ● | 72.69 | 77.58 ● | 72.39 | 77.58 ● | 72.39 | 77.72 ● | 72.28 | 77.69 ● | 72.94 | 77.34 ● |
| hypothyroid | 83.18 | 88.79 ● | 79.58 | 88.72 ● | 83.12 | 88.79 ● | 82.33 | 88.79 ● | 83.18 | 88.80 ● | 88.05 | 89.02 |
| ionosphere | 72.59 | 83.90 ● | 73.85 | 83.88 ● | 73.62 | 83.82 ● | 73.59 | 83.96 ● | 72.56 | 83.96 ● | 75.13 | 85.02 ● |
| iris | 72.00 | 86.87 ● | 72.27 | 87.13 ● | 71.87 | 87.13 ● | 71.80 | 87.07 ● | 72.27 | 86.67 ● | 75.33 | 90.93 ● |
| kr-vs-kp | 76.04 | 85.33 ● | 76.08 | 85.32 ● | 76.06 | 85.32 ● | 76.02 | 85.32 ● | 75.98 | 85.32 ● | 78.10 | 85.61 ● |
| labor | 66.14 | 80.17 ● | 67.37 | 82.98 ● | 63.33 | 80.35 ● | 68.95 | 79.47 ● | 65.26 | 80.70 ● | 50.35 | 81.75 ● |
| letter | 71.18 | 90.58 ● | 72.75 | 91.20 ● | 73.32 | 91.19 ● | 71.20 | 90.57 ● | 71.24 | 90.57 ● | 78.99 | 91.32 ● |
| lymph | 69.26 | 84.80 ● | 69.19 | 84.80 ● | 69.53 | 84.80 ● | 69.66 | 84.80 ● | 69.19 | 84.80 ● | 70.14 | 85.81 ● |
| mushroom | 76.66 | 88.02 ● | 76.67 | 88.02 ● | 76.69 | 88.02 ● | 76.67 | 88.02 ● | 76.70 | 88.02 ● | 79.41 | 88.37 ● |
| segment | 69.64 | 87.63 ● | 69.33 | 87.51 ● | 69.65 | 87.53 ● | 69.67 | 87.64 ● | 69.58 | 87.61 ● | 75.80 | 88.46 ● |
| sick | 81.26 | 84.56 ● | 77.50 | 84.30 ● | 81.20 | 84.54 ● | 80.50 | 84.50 ● | 81.27 | 84.58 ● | 84.93 | 85.12 |
| sonar | 71.54 | 78.56 ● | 72.40 | 79.04 ● | 72.07 | 78.94 ● | 72.36 | 78.80 ● | 71.54 | 78.85 ● | 69.76 | 79.42 ● |
| spambase | 78.36 | 83.40 ● | 78.28 | 83.39 ● | 78.36 | 83.39 ● | 78.33 | 83.40 ● | 78.36 | 83.40 ● | 80.12 | 83.64 ● |
| tic-tac-toe | 76.75 | 77.83 | 75.96 | 77.80 | 76.70 | 77.82 | 76.54 | 77.80 | 76.86 | 77.83 | 78.66 | 76.75 ○ |
| vehicle | 72.67 | 82.45 ● | 72.71 | 82.37 ● | 72.71 | 82.38 ● | 72.72 | 82.46 ● | 72.73 | 82.42 ● | 76.76 | 82.35 ● |
| vote | 76.57 | 83.10 ● | 76.39 | 83.11 ● | 76.50 | 83.11 ● | 76.46 | 83.08 ● | 76.69 | 83.11 ● | 78.80 | 84.09 ● |
| vowel | 73.49 | 91.53 ● | 73.48 | 91.53 ● | 73.47 | 91.53 ● | 73.48 | 91.53 ● | 73.55 | 91.53 ● | 78.23 | 91.79 ● |
| waveform | 74.53 | 84.84 ● | 74.01 | 84.51 ● | 74.45 | 84.73 ● | 74.49 | 84.84 ● | 74.55 | 84.86 ● | 79.05 | 85.73 ● |
| zoo | 74.06 | 87.43 ● | 73.96 | 87.43 ● | 73.57 | 87.53 ● | 73.47 | 87.43 ● | 74.46 | 87.43 ● | 79.80 | 88.62 ● |
| Mean | 74.56 | 83.51 | 74.08 | 83.57 | 74.57 | 83.54 | 74.43 | 83.45 | 74.55 | 83.54 | 76.80 | 83.99 |
| W/T/L | - | 30/4/0 | - | 30/4/0 | - | 30/4/0 | - | 30/4/0 | - | 30/4/0 | - | 27/6/1 |

unchanged, and the results are shown in Table 3. Specifically, Table 3 shows the integration accuracies of six label integration algorithms after label completion by WSLC and LDPLC on the Gaussian distribution and the t-test results. From these results, we can still see that LDPLC further improves the performance of all label integration algorithms compared to WSLC. These results demonstrate that LDPLC is not sensitive to annotation quality distributions.

Besides WSLC, PMF and PMF-TLC have also been proposed for label completion, but they can only handle binary-class crowdsourcing tasks. To further evaluate the effectiveness of LDPLC, we conduct additional experiments to compare the performance of PMF, PMT-TLC and LDPLC on all binary datasets in Table 1. Detailed experiments and results are provided in **Appendix B**. These results also demonstrate the effectiveness of LDPLC.

### 4.4. Parameter Sensitivity Analysis of LDPLC

In LDPLC, we can adjust two parameters: $K$ and $T$. $K$ and $T$ denote the number of neighbors and the number

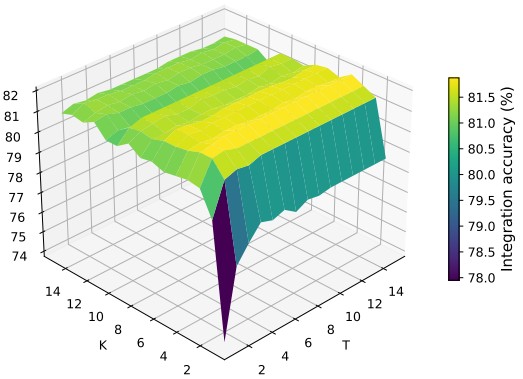

*Figure 4.* Integration accuracy (%) comparisons of MV after label completion by LDPLC on the "LabelMe" dataset when $K$ and $T$ vary from 1 to 15.

of iterations in the label distribution propagation process, respectively. Now, we analyze the impact of these two

*Table 3.* Integration accuracy (%) comparisons of six label integration algorithms after label completion by WSLC and LDPLC on the Gaussian distribution, respectively.

| Dataset | MV WSLC | MV LDPLC | DS WSLC | DS LDPLC | IWMV WSLC | IWMV LDPLC | GTIC WSLC | GTIC LDPLC | DEWSMV WSLC | DEWSMV LDPLC | MNLDP WSLC | MNLDP LDPLC |
|---|---|---|---|---|---|---|---|---|---|---|---|---|
| anneal | 68.30 | 84.37 ● | 65.69 | 84.16 ● | 68.35 | 84.32 ● | 67.59 | 84.21 ● | 68.23 | 84.36 ● | 71.35 | 84.26 ● |
| audiology | 68.54 | 81.33 ● | 68.41 | 81.33 ● | 68.54 | 81.33 ● | 68.54 | 79.42 ● | 68.63 | 81.28 ● | 69.69 | 80.44 ● |
| autos | 69.42 | 81.76 ● | 69.27 | 81.76 ● | 69.37 | 81.80 ● | 69.32 | 81.76 ● | 69.27 | 81.80 ● | 71.22 | 81.12 ● |
| balance-scale | 73.81 | 83.02 ● | 71.31 | 82.24 ● | 73.58 | 82.82 ● | 73.01 | 82.90 ● | 73.73 | 82.99 ● | 78.64 | 83.74 ● |
| biodeg | 78.40 | 81.65 ● | 78.11 | 81.56 ● | 78.37 | 81.59 ● | 78.26 | 81.64 ● | 78.47 | 81.66 ● | 80.26 | 81.83 |
| breast-cancer | 77.45 | 77.55 | 76.61 | 77.52 | 77.20 | 77.52 | 77.03 | 77.55 | 77.31 | 77.52 | 79.37 | 77.59 |
| breast-w | 80.04 | 86.28 ● | 79.89 | 86.15 ● | 80.00 | 86.22 ● | 79.88 | 86.28 ● | 80.10 | 86.32 ● | 83.55 | 89.07 ● |
| car | 79.19 | 87.03 ● | 76.39 | 87.03 ● | 79.23 | 87.04 ● | 78.53 | 87.04 ● | 79.17 | 87.04 ● | 83.89 | 87.24 ● |
| credit-a | 74.81 | 80.19 ● | 75.41 | 80.16 ● | 75.03 | 80.22 ● | 75.04 | 80.23 ● | 74.83 | 80.20 ● | 76.29 | 80.67 ● |
| credit-g | 74.48 | 74.70 | 73.76 | 74.63 | 74.45 | 74.72 | 74.43 | 74.67 | 74.51 | 74.70 | 76.13 | 74.15 |
| diabetes | 76.63 | 77.88 | 75.37 | 77.80 | 76.42 | 77.83 | 76.06 | 77.75 | 76.58 | 77.79 | 78.11 | 77.92 |
| heart-c | 76.24 | 84.29 ● | 75.25 | 84.29 ● | 75.98 | 84.29 ● | 74.85 | 84.29 ● | 76.17 | 84.29 ● | 80.79 | 84.49 ● |
| heart-h | 76.12 | 82.99 ● | 74.76 | 82.99 ● | 75.95 | 82.99 ● | 74.59 | 82.99 ● | 76.05 | 83.02 ● | 80.61 | 83.30 |
| heart-statlog | 73.63 | 78.00 | 73.70 | 77.85 | 73.78 | 77.82 | 73.56 | 78.00 | 73.67 | 78.00 | 74.11 | 77.78 |
| hepatitis | 65.68 | 79.29 ● | 66.58 | 79.61 ● | 67.42 | 79.74 ● | 66.90 | 79.23 ● | 65.68 | 79.22 ● | 61.93 | 79.93 ● |
| horse-colic | 71.66 | 78.12 ● | 72.06 | 77.93 ● | 71.90 | 77.93 ● | 71.79 | 78.04 ● | 71.66 | 78.02 ● | 72.15 | 77.47 ● |
| hypothyroid | 83.16 | 88.49 ● | 79.87 | 88.36 ● | 83.13 | 88.48 ● | 82.48 | 88.47 ● | 83.15 | 88.48 ● | 87.74 | 88.86 |
| ionosphere | 68.20 | 77.95 ● | 70.09 | 78.52 ● | 70.20 | 78.32 ● | 70.17 | 77.89 ● | 68.32 | 77.92 ● | 69.54 | 79.03 ● |
| iris | 71.07 | 87.13 ● | 71.40 | 87.33 ● | 72.40 | 86.87 ● | 72.87 | 87.67 ● | 70.60 | 87.07 ● | 75.53 | 91.47 ● |
| kr-vs-kp | 75.78 | 84.25 ● | 75.77 | 84.24 ● | 75.80 | 84.25 ● | 75.75 | 84.23 ● | 75.81 | 84.23 ● | 77.50 | 84.62 ● |
| labor | 60.88 | 79.30 ● | 62.98 | 79.65 ● | 62.98 | 79.83 ● | 66.32 | 79.12 ● | 60.70 | 78.95 ● | 39.65 | 78.42 ● |
| letter | 71.44 | 90.74 ● | 67.67 | 88.55 ● | 71.87 | 90.64 ● | 71.42 | 90.74 ● | 71.43 | 90.73 ● | 79.52 | 91.53 ● |
| lymph | 69.05 | 83.51 ● | 69.05 | 83.51 ● | 69.39 | 83.45 ● | 69.32 | 83.45 ● | 69.12 | 83.58 ● | 71.28 | 83.85 ● |
| mushroom | 76.96 | 87.85 ● | 76.97 | 87.85 ● | 76.97 | 87.85 ● | 76.96 | 87.85 ● | 76.95 | 87.85 ● | 79.04 | 88.20 ● |
| segment | 70.75 | 87.53 ● | 70.75 | 87.50 ● | 70.75 | 87.48 ● | 70.74 | 87.55 ● | 70.72 | 87.53 ● | 77.20 | 88.59 ● |
| sick | 79.42 | 82.33 | 75.47 | 81.96 ● | 79.40 | 82.36 | 78.50 | 82.26 ● | 79.36 | 82.32 | 82.93 | 82.99 |
| sonar | 69.81 | 78.99 ● | 70.15 | 79.62 ● | 69.62 | 79.52 ● | 70.14 | 79.18 ● | 69.66 | 79.04 ● | 67.89 | 78.94 ● |
| spambase | 76.87 | 81.63 ● | 76.83 | 81.63 ● | 76.87 | 81.62 ● | 76.85 | 81.64 ● | 76.88 | 81.64 ● | 78.43 | 82.01 ● |
| tic-tac-toe | 76.36 | 77.47 | 75.27 | 77.47 | 76.36 | 77.50 | 76.09 | 77.50 | 76.34 | 77.46 | 78.53 | 76.65 ○ |
| vehicle | 72.78 | 81.97 ● | 72.86 | 81.97 ● | 72.86 | 81.93 ● | 72.94 | 81.98 ● | 72.80 | 82.08 ● | 76.50 | 82.17 ● |
| vote | 77.61 | 84.99 ● | 77.49 | 84.99 ● | 77.68 | 85.01 ● | 77.68 | 84.99 ● | 77.75 | 84.99 ● | 79.91 | 86.35 ● |
| vowel | 71.18 | 89.94 ● | 71.22 | 89.94 ● | 71.23 | 89.94 ● | 71.20 | 89.94 ● | 71.28 | 89.94 ● | 75.86 | 90.17 ● |
| waveform | 72.65 | 83.36 ● | 72.22 | 82.98 ● | 72.51 | 83.25 ● | 72.59 | 83.32 ● | 72.56 | 83.35 ● | 77.52 | 84.31 ● |
| zoo | 74.46 | 86.44 ● | 74.56 | 86.44 ● | 73.96 | 86.44 ● | 73.57 | 86.54 ● | 74.46 | 86.44 ● | 79.80 | 87.43 ● |
| Mean | 73.61 | 82.72 | 73.04 | 82.63 | 73.81 | 82.73 | 73.68 | 82.66 | 73.59 | 82.70 | 75.66 | 83.13 |
| W/T/L | - | 28/6/0 | - | 29/5/0 | - | 28/6/0 | - | 29/5/0 | - | 28/6/0 | - | 25/8/1 |

parameters on MV after label completion by LDPLC on the "LabelMe" dataset. We change both $K$ and $T$ from 1 to 15 and set the step size to 1. Figure 4 shows integration accuracies of MV after label completion by LDPLC when $K$ and $T$ vary. We can see that as $K$ varies from 2 to 15, the integration accuracies of all label integration algorithms vary within one percentage point. This indicates that LDPLC is rarely sensitive to $K$. Considering that $K$ takes values in the interval [3, 6] when LDPLC achieves the best performance, we set $K$ to 5 in this paper. As $T$ varies, the integration accuracy increases rapidly at first in the interval $[1, 4]$ and then stabilizes after 4, which is consistent with the analysis in subsection 3.5. Therefore, we set $T$ to 5 in this paper.

## 5. Conclusion

WSLC has been proven to be an effective algorithm for label completion. In this paper, we find that WSLC considers solely the correlation of the labels annotated by different workers on per individual instance while totally ignoring the correlation of the labels annotated by different workers

among similar instances. To address this issue, we proposed a novel label distribution propagation-based label completion (LDPLC) algorithm. In LDPLC, we first use worker similarity weighted majority voting to initialize a label distribution for each missing label. Then, we design a label distribution propagation algorithm to enable each missing label of each instance to iteratively absorb its neighbors' label distributions. Finally, we complete each missing label based on its converged label distribution. Extensive experiments demonstrate that LDPLC further improves the performance of label integration algorithms compared to WSLC.

Nevertheless, there are still some limitations in LDPLC. For example, in crowdsourcing scenarios with high noise ratios, the quality of the labels tends to be low, which results in inaccurately initializing the label distributions. These inaccurately label distributions enlarge the propagation of erroneous information during the label distribution propagation process. Therefore, how to further extend LDPLC to the crowdsourcing scenarios with high noise ratios is the main direction of our future work.

## Acknowledgments

The work was partially supported by National Natural Science Foundation of China (62276241) and Hubei Provincial Collaborative Innovation Center for Basic Education Information Technology Services (OFHUE202312).

## Impact Statement

This paper presents work whose goal is to advance the field of Machine Learning. There are many potential societal consequences of our work, none which we feel must be specifically highlighted here.

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

## A. Experiments on the "Ruters" and "Leaves" datasets.

The dataset "Ruters" contains 1799 instances and 8 classes. In total, there are 5410 multiple noisy labels from 38 workers, with a mean annotation quality of 59.59% and a mean annotation ratio of 7.91%. The dataset "Leaves" contains 384 instances and 6 classes. In total, there are 3840 multiple noisy labels from 83 workers, with a mean annotation quality of 55.03% and a mean annotation ratio of 12.05%. The integration accuracies of six label integration algorithms after label completion by WSLC and LDPLC on the "Ruters" and "Leaves" datasets are shown in Figure 5. From these results, we can conclude that LDPLC further improves the performance of all label integration algorithms compared to WSLC.

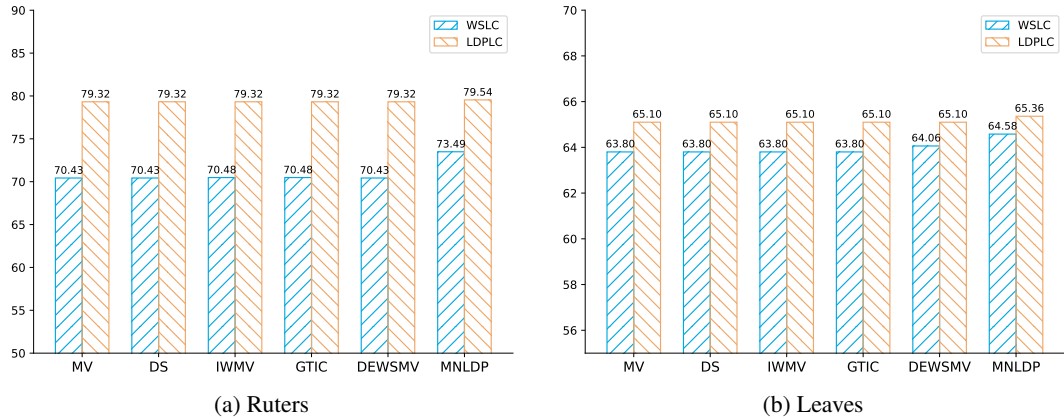

(a) Ruters        (b) Leaves

*Figure 5.* Integration accuracy (%) comparisons of six label integration algorithms after label completion by WSLC and LDPLC, respectively.

## B. Experiments on 18 simulated binary datasets.

We construct two sets of experiments on 18 simulated binary datasets. The first set of experiments compares the performance of PMF and LDPLC, and the second set of experiments compares the performance of PMF-TLC and LDPLC. In each set of experiments, we simulate 40 crowd workers. For each simulated crowd worker, we randomly sample its annotation ratio and annotation quality from the uniform distribution $[0, 0.1]$ and $[0.6, 0.9]$, respectively. The experimental results are shown in Table 4 and Table 5. From the results, we can conclude that LDPLC further improves the performance of most label integration algorithms compared to PMF and PMF-TLC on binary datasets.

*Table 4.* Integration accuracy (%) comparisons of six label integration algorithms after label completion by PMF and LDPLC on 18 simulated binary datasets, respectively.

| | MV | | DS | | IWMV | | GTIC | | DEWSMV | | MNLDP | |
|---|---|---|---|---|---|---|---|---|---|---|---|---|
| Dataset | PMF | LDPLC | PMF | LDPLC | PMF | LDPLC | PMF | LDPLC | PMF | LDPLC | PMF | LDPLC |
| biodeg | 72.48 | 81.55 ● | 74.60 | 81.53 ● | 72.35 | 81.53 ● | 75.77 | 81.56 ● | 74.83 | 81.53 ● | 77.54 | 81.76 ● |
| breast-cancer | 38.15 | 80.66 ● | 41.96 | 80.80 ● | 36.96 | 80.63 ● | 41.53 | 80.73 ● | 38.01 | 80.70 ● | 34.34 | 79.79 ● |
| breast-w | 49.40 | 85.14 ● | 48.27 | 85.04 ● | 43.62 | 85.14 ● | 62.43 | 85.16 ● | 49.03 | 85.21 ● | 50.66 | 87.39 ● |
| credit-a | 58.71 | 82.74 ● | 58.83 | 82.78 ● | 58.55 | 82.75 ● | 63.59 | 82.72 ● | 58.70 | 82.69 ● | 57.55 | 82.74 ● |
| credit-g | 70.62 | 77.28 | 72.77 | 77.26 | 69.92 | 77.31 | 75.96 | 77.24 | 73.21 | 77.27 | 71.75 | 76.99 |
| diabetes | 49.86 | 78.18 ● | 50.56 | 78.20 ● | 46.38 | 78.22 ● | 68.43 | 78.11 ● | 49.88 | 78.18 ● | 49.27 | 78.20 ● |
| heart-statlog | 48.34 | 80.30 ● | 49.89 | 80.11 ● | 47.85 | 80.11 ● | 47.85 | 80.11 ● | 48.26 | 80.26 ● | 46.67 | 79.18 ● |
| hepatitis | 77.48 | 82.52 ● | 76.84 | 82.45 ● | 77.35 | 82.32 ● | 76.12 | 82.32 ● | 77.48 | 82.71 ● | 78.84 | 83.68 ● |
| horse-colic | 63.53 | 79.10 ● | 63.37 | 79.08 ● | 63.64 | 79.08 ● | 64.89 | 79.05 ● | 63.56 | 79.08 ● | 63.56 | 78.75 ● |
| ionosphere | 65.98 | 83.13 ● | 65.33 | 82.79 ● | 65.73 | 82.96 ● | 64.84 | 83.13 ● | 65.93 | 82.96 ● | 65.30 | 84.33 ● |
| kr-vs-kp | 76.93 | 86.50 ● | 77.06 | 86.49 ● | 76.81 | 86.49 ● | 79.22 | 86.50 ● | 77.14 | 86.51 ● | 84.91 | 86.72 ● |
| labor | 66.14 | 80.17 ● | 66.32 | 81.58 ● | 66.32 | 81.93 ● | 65.26 | 79.30 ● | 65.96 | 80.88 ● | 66.84 | 83.68 ● |
| mushroom | 76.46 | 87.90 ● | 76.68 | 87.91 ● | 76.37 | 87.91 ● | 78.92 | 87.91 ● | 76.71 | 87.91 ● | 88.44 | 88.22 |
| sick | 78.26 | 86.51 ● | 80.40 | 86.22 ● | 78.13 | 86.52 ● | 73.66 | 86.47 ● | 82.68 | 86.53 ● | 86.55 | 87.23 |
| sonar | 56.59 | 81.25 ● | 56.73 | 81.73 ● | 56.40 | 81.64 ● | 58.65 | 81.44 ● | 56.64 | 81.39 ● | 55.91 | 81.54 ● |
| spambase | 76.55 | 83.88 ● | 77.66 | 83.88 ● | 76.59 | 83.88 ● | 79.58 | 83.88 ● | 77.77 | 83.88 ● | 83.72 | 84.16 |
| tic-tac-toe | 75.48 | 78.55 ● | 77.37 | 78.57 ● | 75.19 | 78.56 ● | 79.47 | 78.57 | 77.43 | 78.53 ● | 77.91 | 77.77 |
| vote | 43.63 | 86.16 ● | 44.90 | 86.12 ● | 43.01 | 86.12 ● | 45.7 | 86.16 ● | 43.63 | 86.14 ● | 41.82 | 86.83 ● |
| Mean | 63.59 | 82.31 | 64.42 | 82.36 | 62.84 | 82.39 | 66.77 | 82.24 | 64.27 | 82.35 | 65.64 | 82.72 |
| W/T/L | - | 17/1/0 | - | 16/2/0 | - | 17/1/0 | - | 16/2/0 | - | 16/2/0 | - | 13/5/0 |

*Table 5.* Integration accuracy (%) comparisons of six label integration algorithms after label completion by PMF-TLC and LDPLC on 18 simulated binary datasets, respectively.

| Dataset | MV PMF-TLC | MV LDPLC | DS PMF-TLC | DS LDPLC | IWMV PMF-TLC | IWMV LDPLC | GTIC PMF-TLC | GTIC LDPLC | DEWSMV PMF-TLC | DEWSMV LDPLC | MNLDP PMF-TLC | MNLDP LDPLC |
|---|---|---|---|---|---|---|---|---|---|---|---|---|
| biodeg | 74.38 | 81.55 ● | 75.82 | 81.53 ● | 75.21 | 81.53 ● | 78.38 | 81.56 ● | 77.45 | 81.53 ● | 83.08 | 81.76 |
| breast-cancer | 76.43 | 80.66 ● | 70.77 | 80.80 ● | 77.38 | 80.63 ● | 79.79 | 80.73 | 78.74 | 80.70 | 78.32 | 79.79 |
| breast-w | 76.94 | 85.14 ● | 67.98 | 85.04 ● | 77.97 | 85.14 ● | 83.03 | 85.16 ● | 78.76 | 85.21 ● | 93.58 | 87.39 ○ |
| credit-a | 76.09 | 82.74 ● | 57.70 | 82.78 ● | 76.25 | 82.75 ● | 79.91 | 82.72 ● | 75.35 | 82.69 ● | 83.68 | 82.74 |
| credit-g | 73.19 | 77.28 ● | 75.56 | 77.26 | 73.85 | 77.31 ● | 79.60 | 77.24 | 76.46 | 77.27 | 76.72 | 76.99 |
| diabetes | 73.58 | 78.18 ● | 66.19 | 78.20 ● | 74.43 | 78.22 ● | 80.33 | 78.11 ○ | 75.44 | 78.18 ● | 78.67 | 78.20 |
| heart-statlog | 73.63 | 80.30 ● | 58.81 | 80.11 ● | 74.74 | 80.11 ● | 75.77 | 80.11 ● | 74.85 | 80.26 ● | 80.78 | 79.18 |
| hepatitis | 76.84 | 82.52 ● | 80.71 | 82.45 | 75.68 | 82.32 ● | 73.16 | 82.32 ● | 73.61 | 82.71 ● | 85.68 | 83.68 |
| horse-colic | 72.42 | 79.10 ● | 63.94 | 79.08 ● | 72.47 | 79.08 ● | 78.09 | 79.05 | 71.22 | 79.08 ● | 79.29 | 78.75 |
| ionosphere | 74.84 | 83.13 ● | 66.38 | 82.79 ● | 75.84 | 82.96 ● | 76.23 | 83.13 ● | 73.65 | 82.96 ● | 84.62 | 84.33 |
| kr-vs-kp | 80.47 | 86.50 ● | 80.86 | 86.49 ● | 80.45 | 86.49 ● | 83.08 | 86.50 ● | 80.74 | 86.51 ● | 88.92 | 86.72 ○ |
| labor | 77.37 | 80.17 | 55.09 | 81.58 ● | 76.14 | 81.93 ● | 74.03 | 79.30 ● | 76.32 | 80.88 | 84.74 | 83.68 |
| mushroom | 81.27 | 87.90 ● | 81.54 | 87.91 ● | 81.33 | 87.91 ● | 83.89 | 87.91 ● | 81.48 | 87.91 ● | 92.08 | 88.22 ○ |
| sick | 83.24 | 86.51 ● | 88.36 | 86.22 ○ | 83.29 | 86.52 ● | 81.99 | 86.47 ● | 88.17 | 86.53 | 91.45 | 87.23 ○ |
| sonar | 73.41 | 81.25 ● | 56.11 | 81.73 ● | 73.89 | 81.64 ● | 78.07 | 81.44 ● | 74.04 | 81.39 ● | 83.17 | 81.54 |
| spambase | 78.19 | 83.88 ● | 79.48 | 83.88 ● | 78.00 | 83.88 ● | 80.65 | 83.88 ● | 79.29 | 83.88 ● | 85.57 | 84.16 ○ |
| tic-tac-toe | 74.92 | 78.55 ● | 75.90 | 78.57 | 75.88 | 78.56 ● | 79.72 | 78.57 | 77.28 | 78.53 | 72.89 | 77.77 ● |
| vote | 78.23 | 86.16 ● | 63.61 | 86.12 ● | 77.81 | 86.12 ● | 84.18 | 86.16 ● | 79.22 | 86.14 ● | 91.40 | 86.83 ○ |
| Mean | 76.41 | 82.31 | 70.27 | 82.36 | 76.70 | 82.39 | 79.45 | 82.24 | 77.34 | 82.35 | 84.15 | 82.72 |
| W/T/L | - | 17/1/0 | - | 14/3/1 | - | 18/0/0 | - | 13/4/1 | - | 13/5/0 | - | 1/11/6 |

