# OpenReview forum: "Label Distribution Propagation-based Label Completion for Crowdsourcing"
_ICML.cc/2025/Conference — ICML 2025 poster_

### Official Review · Reviewer_CJ1M · 2025-02-24

**Overall Recommendation:** 4

**Summary:**

To complete the missing labels, this paper proposes a novel label completion method for crowdsourcing by utilizing label distribution propagation. Both the worker similarity and the label correlation are considered to generate the label distribution for missing labels. Based on the worker similarity, the weighted majority voting is applied to obtain the initialized label distribution, and local linear embedding is adopted to finish the label distribution propagation. Experimental results and related analysis demonstrate the effectiveness of the proposal.

## update after rebuttal
The author's response has addressed my concerns, and after reviewing the other reviewers' comments, I support the acceptance of this paper. Therefore, I maintain my original score.

**Claims And Evidence:**

Yes.

**Essential References Not Discussed:**

No.

**Experimental Designs Or Analyses:**

Yes. This paper designs the related experiments to show the effectiveness of the proposal.

**Methods And Evaluation Criteria:**

Yes.

**Other Comments Or Suggestions:**

Regarding the weaknesses of WSLC, the paper only provides a conclusion without conducting a detailed analysis. It is necessary to further analyze the reasons why WSLC has these weaknesses in the introduction.

**Other Strengths And Weaknesses:**

Strengths:
1 This paper studies the label completion problem in crowdsourcing area, which is a very important issue as usually the missing proportion is very high in real-world scenarios.
2 This paper proposes a label completion method based on label distribution propagation. Compared to existing classical WSLC method, both worker similarity and label correlation are considered.
3 This paper theoretically analysis the convergence of the proposed method.
4 This paper is well-written and easy to read.

Weaknesses:
1 There are several label completion methods mentioned in related work, why only compare WSLC only in the experiments?
2 For the methods employed in this paper, it is better to give the thorough analysis and discussion, such as explaining why Pearson correlation is selected to learn feature vectors rather than other correlation methods. Why is cosine similarity employed to obtain worker similarity? An introduction and discussion on the reasonableness of using these methods should be presented.

**Questions For Authors:**

1 There are several label completion methods mentioned in related work, why only compare WSLC only in the experiments?
2 For the methods employed in this paper, it is better to give the thorough analysis and discussion, such as explaining why Pearson correlation is selected to learn feature vectors rather than other correlation methods. Why is cosine similarity employed to obtain worker similarity?

**Relation To Broader Scientific Literature:**

This paper discusses the effectiveness of label completion for crowdsourcing problem, and proposes a label propagation method for label completion.

**Theoretical Claims:**

Yes. The theoretical claim is about the convergence of the proposed method.

---

> ### Author Rebuttal · Authors · 2025-04-01
>
> Thanks a lot for your comments. Please find our detailed responses to your concerns as follows.
>
> **Author Response to Q1:** We choose WSLC as the primary comparison method because WSLC is the most recent and relevant for our work. Moreover, both WSLC and our work are designed to address multi-class crowdsourcing tasks. In contrast, PMF and PMF-TLC are primarily designed for binary-class crowdsourcing tasks. To address the reviewer’s concerns, we conduct additional comparative experiments on four simulated datasets specifically designed for binary-class crowdsourcing tasks. The results are as follows:
>
> |Dataset|MV|GTIC|DEWSMV|MNLDP|
> |--|--|--|--|--|
> |biodeg (PMF)|73.18%|75.77%|76.30%|78.39%|
> |biodeg (PMF-TLC)|74.22%|78.38%|77.73%|**80.56%**|
> |biodeg (LDPLC)|**79.05%**|**81.36%**|**79.05%**|79.43%|
> |breast-w (PMF)|54.36%|62.43%|55.94%|51.36%|
> |breast-w (PMF-TLC)|77.83%|83.03%|79.54%|85.99%|
> |breast-w (LDPLC)|**83.12%**|**85.97%**|**83.12%**|**86.98%**|
> |credit-a (PMF)|59.57%|63.59%|59.28%|57.83%|
> |credit-a (PMF-TLC)|73.62%|79.91%|74.78%|82.03%|
> |credit-a (LDPLC)|**82.90%**|**80.48%**|**82.90%**|**83.19%**|
> |diabetes (PMF)|41.93%|68.43%|43.88%|37.89%|
> |diabetes (PMF-TLC)|74.87%|**80.43%**|77.08%|79.08%|
> |diabetes (LDPLC)|**79.04%**|79.18%|**79.04%**|**79.56%**|
> ||
>
> From the experimental results, it can be observed that LDPLC outperforms PMF and PMF-TLC in most binary-class crowdsourcing tasks, further demonstrating its effectiveness and applicability in label completion tasks. In the final version of the paper, we will add the reasons why we only compare WSLC in the experiments.
>
> **Author Response to Q2:** To address the reviewer's concerns, we conduct two groups of experiments: (1) We compare the performance of Pearson correlation and mutual information for learning workers' feature vectors. We refer to the method with mutual information as LDPLC-MI. (2) We compare the performance of cosine similarity and Euclidean distance for estimating worker similarity using the learned feature vectors. We refer to the method with Euclidean distance as LDPLC-ED.
>
> We conduct the first group of experiments on all seven simulated datasets with discrete variables. The results are as follows:
> |Dataset|MV|GTIC|DEWSMV|MNLDP|
> |--|--|--|--|--|
> |audiology (LDPLC-MI)|80.53%|79.65%|80.53%|78.32%|
> |audiology (LDPLC)|80.53%|79.65%|80.53%|78.32%|
> |breast-cancer (LDPLC-MI)|74.48%|74.48%|74.48%|71.68%|
> |breast-cancer (LDPLC)|**75.17%**|**75.17%**|**75.52%**|**75.52%**|
> |car (LDPLC-MI)|83.56%|83.56%|83.56%|**85.76%**|
> |car (LDPLC)|83.56%|83.56%|83.56%|85.65%|
> |kr-vs-kp (LDPLC-MI)|**85.54%**|**85.51%**|**85.51%**|**85.70%**|
> |kr-vs-kp (LDPLC)|85.45%|85.42%|85.45%|85.64%|
> |mushroom (LDPLC-MI)|**91.31%**|**91.30%**|91.30%|91.49%|
> |mushroom (LDPLC)|91.30%|91.29%|91.30%|91.49%|
> |tic-tac-toe (LDPLC-MI)|**81.63%**|**81.63%**|**81.63%**|80.48%|
> |tic-tac-toe (LDPLC)|81.52%|81.42%|81.42%|**80.69%**|
> |vote (LDPLC-MI)|82.53%|82.53%|82.53%|84.83%|
> |vote (LDPLC)|**82.99%**|**82.99%**|**82.99%**|**85.06%**|
> ||
>
> From the results, it can be found that there is no significant difference in performance between Pearson correlation and mutual information. However, it's worth noting that mutual information can only be used to measure the correlation of discrete variables. With the aim of broadening the applicability of the proposed LDPLC, we choose Pearson correlation as the preferred method to learn workers' feature vectors.
>
> We conduct the second group of comparative experiments on the "LabelMe" datasets. The results are as follows:
>
> |Dataset|MV|GTIC|DEWSMV|MNLDP|
> |--|--|--|--|--|
> |LabelMe (LDPLC-ED)|81.4%|81.4%|81.4%|81.7%|
> |LabelMe (LDPLC)|**81.7%**|**81.7%**|**81.6%**|**82.5%**|
> ||
>
> The experimental results clearly indicate that cosine similarity is better than Euclidean distance in estimating worker similarity. Therefore, we choose cosine similarity as the preferred method to estimate worker similarity. In the final version of the paper, we will give related analysis and discussion of the methods employed in this paper.
>
> **Author Response to WSLC's Weaknesses:** The core assumption of WSLC is that workers with similar cognitive abilities will annotate similar labels on the same instances. Therefore, it completes each missing label solely based on the labels annotated by similar workers on this corresponding instance. However, in real-world crowdsourcing scenarios, each instance usually has few labels, thus WSLC struggles to complete its missing labels. By introducing the assumption that the same worker will also annotate similar labels on similar instances, the missing labels can not only be inferred from labels of similar workers on the same instance but also absorb the distribution information of labels from all workers across neighboring instances. In the final version of the paper, we will further analyze the reasons why WSLC has these weaknesses in the introduction.

---

### Official Review · Reviewer_vQCd · 2025-03-06

**Overall Recommendation:** 4

**Summary:**

This paper proposes a novel label distribution propagation-based label completion (LDPLC) algorithm to address the sparsity issue in crowdsourced label matrices. Existing worker similarity-based label completion (WSLC) algorithm only considers the correlation of labels annotated by different workers on individual instances, ignoring the correlation of the labels annotated by different workers among similar instances. To fill this gap, LDPLC initializes label distributions using worker similarity weighted majority voting and then propagates these distributions iteratively to absorb information from neighboring instances. Finally, LDPLC completes each missing label based on converged label distributions. Experimental results on both real-world and simulated datasets validate the effectiveness of LDPLC.

**Claims And Evidence:**

The most important claim made in the paper is that the proposed LDPLC can considers not only the correlation of the labels annotated by different workers on per individual instance, but also the correlation of the labels annotated by different workers among similar instances. By doing so, LDPLC can further improve the performance of label completion.

To support this claim, LDPLC initializes label distributions through worker similarity weighted majority voting, which utilizes the correlation of the labels annotated by different workers on per individual instance. Subsequently, LDPLC iteratively propagates these distributions to absorb information from neighboring instances, which utilizes the correlation of the labels annotated by different workers among similar instances. The results shown in Figure 3, Table 1, and Table 2 demonstrate that LDPLC can further improve the performance of label completion.

**Essential References Not Discussed:**

No. All essential references have been cited/discussed in the paper.

**Experimental Designs Or Analyses:**

Yes. The paper conduct extensive experiments on both real-world and simulated datasets, providing strong empirical evidence to support the effectiveness of LDPLC.

**Methods And Evaluation Criteria:**

Yes. LDPLC utilizes the correlation of the labels annotated by different workers among similar instances through label distribution propagation, which effectively fills the gap left by WSLC. Regarding evaluation criteria, this paper adopts the integration accuracy, which is commonly used in other label completion studies.

**Other Comments Or Suggestions:**

I have found a small issue in writing as follows:
1) On line 107 in page 2, the full name of the “EM” framework should be provided when it first appears.

**Other Strengths And Weaknesses:**

Strengths:
1) This paper reveals a critical limitation of existing work: WSLC considers solely the correlation of the labels annotated by different workers on per individual instance while totally ignoring the correlation of the labels annotated by different workers among similar instances.
2) To fill this gap, this paper proposes a label distribution propagation-based label completion (LDPLC) algorithm. The proposed LDPLC utilizes the correlation of the labels annotated by different workers among similar instances through label distribution propagation.
3) This paper provides a detailed theoretical analysis of the convergence properties of LDPLC (subsection 3.4), ensuring the algorithm's reliability.
4) This paper provides extensive experiments on both real-world and simulated datasets, providing strong empirical evidence to support the effectiveness of LDPLC. The results show that LDPLC significantly outperforms WSLC in terms of label integration accuracy across multiple datasets and label integration algorithms.

Weaknesses:
1) The key motivation for proposing LDPLC in this paper lies in the limitation that WSLC does not take into account the correlation of the labels annotated by different workers among similar instances. However, why WSLC has this limitation and what consequences it may lead to are not discussed. To further clarify the motivation of this paper, these aspects are necessary to address.
2) As shown in Tables 1 and 2, the proposed LDPLC performs significantly worse than WSLC only on the “tic-tac-toe” dataset when the label integration algorithm is MNLDP. Why does this happen? A detailed explanation is necessary.

**Questions For Authors:**

See the Weaknesses parts:
1) The key motivation for proposing LDPLC in this paper lies in the limitation that WSLC does not take into account the correlation of the labels annotated by different workers among similar instances. However, why WSLC has this limitation and what consequences it may lead to are not discussed. To further clarify the motivation of this paper, these aspects are necessary to address.
2) As shown in Tables 1 and 2, the proposed LDPLC performs significantly worse than WSLC only on the “tic-tac-toe” dataset when the label integration algorithm is MNLDP. Why does this happen? A detailed explanation is necessary.

**Relation To Broader Scientific Literature:**

This paper makes a contribution to the field of label completion for crowdsourcing by addressing a critical limitation of existing work. The proposed LDPLC utilizes the correlation of the labels annotated by different workers among similar instances through label distribution propagation, which effectively fills the gap left by existing work and thus further improve the performance of label completion.

**Theoretical Claims:**

Yes. The paper provides a detailed theoretical analysis of the convergence properties of LDPLC. I have verified the correctness of the theoretical analysis.

---

> ### Author Rebuttal · Authors · 2025-03-31
>
> **Q1:** The key motivation for proposing LDPLC in this paper lies in the limitation that WSLC does not take into account the correlation of the labels annotated by different workers among similar instances. However, why WSLC has this limitation and what consequences it may lead to are not discussed. To further clarify the motivation of this paper, these aspects are necessary to address.
>
> **Author Response to Q1:** Thanks for your valuable comments. The core assumption of WSLC is that workers with similar cognitive abilities will annotate similar labels on the same instances. Therefore, it completes each missing label solely based on the labels annotated by similar workers on this corresponding instance. However, in real-world crowdsourcing scenarios, each instance usually has few labels, thus WSLC struggles to complete its missing labels. By introducing the assumption that the same worker will also annotate similar labels on similar instances, the missing labels can not only be inferred from labels of similar workers on the same instance but also absorb the distribution information of labels from all workers across neighboring instances. In the final version of the paper, we will include these discussions on why WSLC has this limitation and what consequences it may lead to. Thanks again for your valuable comments.
>
> **Q2:** As shown in Tables 1 and 2, the proposed LDPLC performs significantly worse than WSLC only on the “tic-tac-toe” dataset when the label integration algorithm is MNLDP. Why does this happen? A detailed explanation is necessary.
>
> **Author Response to Q2:** Thanks for your valuable comments. Through an in-depth understanding of the “tic-tac-toe” dataset, we found that this dataset has a unique feature structure. This dataset encodes the complete set of possible board configurations at the end of tic-tac-toe games. Each instance contains 9 structured features, corresponding to the nine positions on a 3×3 game board, precisely recording the occupancy state of each cell: player X (marked as ‘x’), opponent O (‘o’), or unoccupied (‘b’). The different occupancy state of one cell on the board can directly affect the outcome of the game. However, when corresponding to instances, we consider them similar because only a small number of features differ, leading to incorrect information being propagated across these similar instance. Since MNLDP also has a propagation stage, it propagates more erroneous information, which leads to a decrease in integration accuracy. In the final version of the paper, we will include these discussions on why the proposed LDPLC performs significantly worse than WSLC only on the “tic-tac-toe” dataset when the label integration algorithm is MNLDP. Thanks again for your valuable comments.
>
> **Other Comments Or Suggestions:** On line 107 in page 2, the full name of the “EM” framework should be provided when it first appears.
>
> **Author Response:** Thanks for your valuable comments. In the final version of the paper, we will provide the full name of the EM, i.e., Expectation-Maximization. Thanks again.

---

> > ### Comment · Reviewer_vQCd · 2025-04-07
> >
> > Thank you for the author's response. After considering the other reviewers' comments, I have decided to maintain my original rating.

---

### Official Review · Reviewer_8VZy · 2025-03-10

**Overall Recommendation:** 2

**Summary:**

1. The paper primarily addresses the shortcomings in WSLC, which traditionally considers only the correlations among labels annotated by different workers for individual instances. The authors propose the LDPLC algorithm, which additionally accounts for correlations among labels annotated by different workers across similar instances.
2. Overall, the reported results demonstrate superior performance, consistently outperforming most existing methods on both real-world and simulated datasets.
3. The robustness of the proposed algorithm is well-supported by thorough theoretical analysis and comprehensive parameter sensitivity analysis.

**Claims And Evidence:**

The method proposed in the paper was verified in the experimental stage.

**Essential References Not Discussed:**

The core contribution of this paper is its consideration that different workers with similar cognitive abilities tend to annotate similar labels for the same instance, and that the same worker tends to annotate similar labels across similar instances. In other words, the authors address manifold consistency among workers and labels, as well as among instances and labels. Therefore, I suggest that the paper should include discussions on relevant literature related to manifold consistency.

**Experimental Designs Or Analyses:**

1. The experimental design is comprehensive, including extensive comparisons across multiple datasets and thorough parameter analysis.
2. A potential limitation is the absence of detailed descriptions of dataset characteristics. Therefore, it remains unclear whether the proposed method can perform equally well in scenarios involving a large number of categories.

**Methods And Evaluation Criteria:**

The choice of comparison methods, evaluation criteria, and datasets is appropriate and well-justified.

**Other Comments Or Suggestions:**

The text in Figures 1 and 3 is too small, reducing readability. I suggest enlarging the font size or redesigning these figures to enhance clarity.

**Other Strengths And Weaknesses:**

Strengths:
(1) The paper considers multiple manifold consistency issues within the crowdsourcing learning process and employs local linear embedding for Label Distribution Propagation.
(2) Experiments conducted across various datasets demonstrate the feasibility and effectiveness of the proposed method.
(3) The paper also provides a convergence analysis.

Weaknesses:
(1) The current work is limited to linear scenarios. It would be valuable to discuss whether this approach can be extended to kernel-based methods or deep neural networks.
(2) The authors have not discussed the computational complexity of the algorithm. Specifically, computing neighbors in feature space can be time-consuming, which raises my concerns about practical applicability to real-world scenarios.

**Questions For Authors:**

1. In practical experiments (rather than theoretical analysis alone), what is the convergence efficiency of this algorithm?
2. Can the proposed approach be extended to more complex scenarios, such as kernel-based methods or deep neural networks? This is particularly relevant because, in practice, feature data often do not satisfy linear assumptions.
3. Could the authors discuss the computational complexity of the proposed algorithm?

If the authors address these questions clearly, I am willing to raise my score.


## update after rebuttal
The author's response has partially addressed my concerns, and after reviewing the other reviewers' comments, I think the current article should be supplemented with a kernel-based extension method to enrich the completeness of the article, which is not difficult. Therefore, I maintain my original score.

**Relation To Broader Scientific Literature:**

This paper integrates manifold consistency with crowdsourcing learning.

**Theoretical Claims:**

I am not an expert in this field, but upon reviewing the theoretical claims briefly, I found no obvious errors.

One minor limitation is the absence of a consistency analysis for other architectures (e.g., kernel methods or deep neural networks).

---

> ### Author Rebuttal · Authors · 2025-04-01
>
> Thanks a lot for your comments. Please find our detailed responses to your concerns as follows.
>
> **Author Response to Convergence Efficiency:** As shown in Figure 4, LDPLC converges after just 4 iterations on the “LabelMe” dataset. To address the reviewer's concerns, we further observe the convergence efficiency of LDPLC on the first five simulated datasets. We observe that integration accuracies (%) of MV after label completion by LDPLC when T varies as follows:
>
> |Dataset\T|1|2|3|4|5|6|7|8|9|10|11|12|13|14|15|
> |--|--|--|--|--|--|--|--|--|--|--|--|--|--|--|--|
> |anneal|89.64|94.65|94.99|95.66|95.55|95.55|95.55|95.55|95.55|95.55|95.55|95.55|95.55|95.55|95.55|
> |audiology|89.38|93.36|94.69|94.25|94.69|94.25|94.69|94.25|94.25|94.25|94.25|94.25|94.25|94.25|94.25|
> |autos|88.29|84.88|90.73|90.73|90.73|91.22|91.22|91.22|91.22|91.22|91.22|91.22|91.22|91.22|91.22|
> |balance-scale|76.80|87.20|86.72|87.52|87.04|87.36|87.68|87.36|87.36|87.36|87.36|87.52|87.20|87.52|87.20|
> |biodeg|70.14|72.42|72.80|72.32|71.94|72.23|72.32|72.13|72.13|72.23|71.85|71.85|72.13|72.13|72.23|
> ||
>
> These results confirm that LDPLC can achieve convergence within 10 iterations. In the final version of the paper, we will add the convergence efficiency analysis of LDPLC.
>
> **Author Response to Extension:** To the best of our knowledge, local linear embedding (LLE) is a nonlinear spectral dimensionality reduction and manifold learning method [1]. Because for most datasets, the assumption of local linear correlation in LLE is satisfied. This is why LDPLC performs well on most datasets. However, if the local space is particularly complex, the assumption of LLE may not be satisfied. Therefore, to handle more complex nonlinear problems, we think LDPLC can be extended to more complex scenarios, such as kernel-based methods or deep neural networks. By leveraging kernel methods, such as Kernel LLE, the nonlinear feature information can be extracted when mapping input data into some high dimensional feature space. Moreover, deep neural networks can map a nonlinear feature space to a linear feature space. In the new feature space, we can better find neighbors and optimize their weights. These will become important directions for our future work.
>
> [1] Theoretical Connection between Locally Linear Embedding, Factor Analysis, and Probabilistic PCA;
>
> **Author Response to Computational Complexity:** Below, we provide a detailed analysis of LDPLC’s computational complexity. In Algorithm 1, lines 3-8 learn feature vectors with a time complexity of $O(R(N+M(n_ln_a|D_r|)))$, where $n_l$ and $n_a$ are the average number of values for a label variable and an original feature variable, respectively. Lines 9-13 estimate worker similarity with a time complexity of $O(R^2M)$. Lines 14-22 initialize label distributions with a time complexity of $O(NRQ)$. Lines 23-26 identify neighbors and optimize their weights with a time complexity of $O(N^2M+NK^3)$. Lines 27-29 propagate the label distributions with a time complexity of $O(TNKQ)$. Finally, lines 30-36 complete missing labels with a time complexity of $O(NRQ)$. If only the highest order terms are taken, the overall time complexity of LDPLC is $O(RM(n_ln_a|D_r|)+R^2M+NRQ+N^2M+NK^3+TNKQ)$. In addition, we compare the run time of the WSLC and LDPLC, as follows:
> |Dataset|WSLC|LDPLC|
> |--|--|--|
> |LabelMe|2.48s|12.40s|
> ||
>
> The experimental results show that LDPLC takes more time than WSLC. Although LDPLC slightly increases the run time, it has better performance. In the final version of the paper, we will add the computational complexity of LDPLC.
>
> **Author Response to A Large Number of Categories:** Among 34 simulated datasets, “audiology” and “letter” contain 24 and 26 categories, respectively. In addition, to further address the reviewer’s concerns, we simulate a dataset with 10000 instances and 100 categories, named “Simulation_100”, to conduct a new experiment. The results are as follows:
> |Dataset|MV_WSLC|MV_LDPLC|GTIC_WSLC|GTIC_LDPLC|DEWSMV_WSLC|DEWSMV_LDPLC|MNLDP_WSLC|MNLDP_LDPLC|
> |--|--|--|--|--|--|--|--|--|
> |Simulation_100|90.50%|**92.70%**|90.50%|**92.66%**|90.52%|**92.64%**|91.56%|**92.24%**|
> ||
>
> From these results, it can be found that the integration accuracy of each integration algorithm improves significantly after label completion using LDPLC compared to WSLC. These results further verify that LDPLC can perform equally well in scenarios involving a large number of categories. In the final version of the paper, we will include more detailed descriptions of dataset characteristics.
>
> **Author Response to Essential References:** In the final version of the paper, we will add a discussion of the relevant literature related to manifold consistency.
>
> **Author Response to Figures 1 and 3:** Indeed, the text in Figures 1 and 3 is too small, reducing readability. In the final version of the paper, we will enlarge the font size and optimize the layout of the figures and tables.

---

### Official Review · Reviewer_qna3 · 2025-03-14

**Overall Recommendation:** 3

**Summary:**

This paper proposes a crowdsourcing label completion method to complement subsequent truth-inference/label-integration methods.
The proposed method primarily focuses on improving the existing method WSLC, which “considers solely the correlation of the labels annotated by different workers on per individual instance while totally ignoring the correlation of the labels annotated by different workers among similar instances”.

## update after rebuttal

Thanks for the feedback from the authors. I have raised my scores accordingly.

**Claims And Evidence:**

Regarding this point, my main concern lies in the experiments. Please refer to the reviews on “Experimental Designs Or Analyses”.

**Essential References Not Discussed:**

It is recommended to cite Reference [1].
[1] Truth inference in crowdsourcing: Is the problem solved? VLDB 2017.

**Experimental Designs Or Analyses:**

The experiment section has the following mian issues:
1) Only one real-world dataset, LabelMe, is used. In fact, there are many real-world crowdsourcing annotation datasets available, e.g., [1, 2].
2) The truth inference methods considered lack the classic DS method and other methods. It is recommended to include the DS method and the methods used in benchmark [1].
3) The experimental analysis is insufficient. In summary, for experiments on real-world datasets, all results are only involes Figure 3.

[1] Truth inference in crowdsourcing: Is the problem solved? VLDB 2017.
[2] Deep learning from crowds. AAAI 2018.

**Methods And Evaluation Criteria:**

Regarding this point, my main concern lies in the experiments. Please refer to the reviews on “Experimental Designs Or Analyses”.

**Other Comments Or Suggestions:**

none.

**Other Strengths And Weaknesses:**

Strengths:
1)	First, the overall narrative of the paper is clear and easy to follow, with logical flow (especially in the introduction section). The paper has not excessive embellishment.
2)	In the experiment section, particularly on the real-world dataset LabelMe, the proposed method outperforms the optimized method WSLC.

Weaknesses
1）	Regarding the research motivation and the research problem.
The research motivation and the research problem are clearly introduced, but the significance of the addressed problem is not sufficiently compelling.
This is because the paper primarily focuses on improving the existing method WSLC to eliminate its limitation—it “considers solely the correlation of the labels annotated by different workers on per individual instance while totally ignoring the correlation of the labels annotated by different workers among similar instances”.
Furthermore, the proposed method aims to enhance the accuracy of subsequent truth inference processes.
However, the problem of truth inference for crowdsourced annotations is a long-standing issue, with mature benchmark datasets and review papers already available since 2017, e.g., [1].
As a long-standing research problem, publishing high-level papers on truth inference requires higher standards and the resolution of sufficiently significant pain points.

2）	Regarding the experiments.

 Please refer to the above reviews on “Experimental Designs Or Analyses”.


[1] Truth inference in crowdsourcing: Is the problem solved? VLDB 2017.

**Questions For Authors:**

None.

**Relation To Broader Scientific Literature:**

The key contributions of the paper are important for research on supervised learning.

**Theoretical Claims:**

Yes, I have reviewed the methodology part of this paper.

---

> ### Author Rebuttal · Authors · 2025-04-01
>
> Thanks a lot for your comments. Please find our detailed responses to your concerns as follows.
>
> **Author Response to Research Motivation and Research Problem:** Label integration (truth inference) is indeed a long-standing research problem. Over the past decades, numerous algorithms have been proposed to improve the performance of label integration. These algorithms have gradually reached a consensus: when worker annotation is more accurate than random annotation, the more noisy labels an instance receives, the easier it becomes to infer its unknown true label. However, in real-world scenarios, each worker typically annotates only a small number of instances, and few labels are typically collected per instance to reduce cost, resulting in a highly sparse crowdsourced label matrix. This fact leads to label integration failing to achieve the expected performance relying solely on the existing labels in the label matrix. To address this issue, label completion has been proposed to fill in missing labels in sparse label matrices and is gaining increasing attention. To the best of our knowledge, WSLC is currently the most advanced and convincing benchmark for label completion. However, WSLC solely considers the correlation of the labels annotated by different workers on per individual instance. Our proposed algorithm, LDPLC, is the first to jointly model both worker similarity and instance similarity. Specifically, LDPLC first estimates worker similarity by learning feature vectors, enabling a more accurate initialization of label distributions. Next, it finds neighbors and employs locally linear embedding (LLE) to optimize their weights, capturing the geometric structure among instances. Finally, LDPLC propagates label distributions across instances, allowing missing labels to absorb information from similar instances, thereby overcoming the limitations of WSLC.
>
> **Author Response to Experimental Designs:** We have carefully considered these issues and conducted additional experiments to support our work. Below are the updates we make: (1) In addition to “LabelMe”, we have now included two real-world crowdsourced datasets, “Ruters” and “Leaves”.“Ruters” and “Leaves” are also collected from the AMT platform. “Ruters” contains 1799 instances, 5410 labels, 8 classes, and 38 workers. “Leaves” contains 384 instances, 3840 labels, 6 classes, and 83 workers. (2) We have added the DS, KOS, and IWMV algorithms to the experiments to provide a more comprehensive comparative analysis. The experimental results are as follows:
>
> |Dataset|MV_WSLC|MV_LDPLC|GTIC_WSLC|GTIC_LDPLC|DEWSMV_WSLC|DEWSMV_LDPLC|MNLDP_WSLC|MNLDP_LDPLC|DS_WSLC|DS_LDPLC|KOS_WSLC|KOS_LDPLC|IWMV_WSLC|IWMV_LDPLC|
> |--|--|--|--|--|--|--|--|--|--|--|--|--|--|--|
> |LabelMe|76.40%|**81.70%**|76.40%|**81.70%**|76.60%|**81.60%**|80.50%|**82.50%**|76.70%|**81.50%**|76.30%|**81.70%**|76.30%|**81.70%**|
> |Ruters|70.37%|**79.32%**|70.48%|**79.32%**|70.37%|**79.32%**|73.49%|**79.54%**|70.43%|**79.32%**|70.37%|**79.32%**|70.48%|**79.32%**|
> |Leaves|63.80%|**65.10%**|63.80%|**65.10%**|63.80%|**65.10%**|64.58%|**65.36%**|63.80%|**65.10%**|64.06%|**65.10%**|63.80%|**65.10%**|
> ||
>
> From these results, it can be found that the integration accuracies of both classic algorithms (MV, DS, KOS) and the lastest algorithms (GTIC, IWMV, DEWSMV, MNLDP) improve significantly after label completion using LDPLC compared to WSLC across all three datasets. These findings confirm that LDPLC approximates crowds more accurately than WSLC and further improves the performance of all label integration algorithms. These results once again verify the effectiveness and robustness of LDPLC. In the final version of the paper, we will further extend the experiment and analysis on real-world datasets.
>
> **Author Response to Essential References:** By studying this reference, we have gained a clearer understanding of the differences between various classic algorithms and the future directions of crowdsourcing research, which is highly beneficial for our future work. In the final version of the paper, we will cite [1] and discuss its relationship with our work.

---

### Decision · Program_Chairs · 2025-05-01

**Decision:**

Accept (poster)

**Comment:**

This paper tackles an important problem in crowdsourcing, that is, how to fill in missing labels when only a few workers label each item.
It points out a clear weakness in a well-known method (WSLC) and proposes a new solution (LDPLC) that uses both worker similarity and instance similarity. The effectiveness of the proposed method was supported by experiments on a wide range of datasets.

The reviewers raised some concerns, but the authors addressed them carefully with new experiments and detailed explanations. Most reviewers were satisfied with the responses and supported acceptance.

Overall, this paper is well-structured, technically sound, and makes a clear contribution. It meets the standard for acceptance.